

# Modulating factors of the abundance and distribution of *Achelous spinimanus* (Latreille, 1819) (Decapoda, Portunoidea), a fishery resource, in Southeastern Brazil

Aline Nonato de Sousa[1], Giovana Bertini[1,2], Fabiano Gazzi Taddei[1,3], Rogério Caetano da Costa[1,4], Thiago Elias da Silva[1] and Adilson Fransozo[1]

[1] Núcleo de Estudos em Biologia, Ecologia e Cultivo de Crustáceos—NEBECC, Universidade Estadual Paulista "Júlio de Mesquita Filho"—UNESP, Botucatu, São Paulo, Brasil

[2] Laboratório de Biologia e Cultivo de Crustáceos—LABCRUST, Universidade Estadual Paulista "Júlio de Mesquita Filho"—UNESP, Registro, São Paulo, Brasil

[3] Laboratório de Estudos de Crustáceos Amazônicos—LECAM, Universidade do Estado do Amazonas—UEA, Parintins, Amazonas, Brasil

[4] Laboratório de Biologia de Camarões Marinhos e de Água Doce—LABCAM, Universidade Estadual Paulista "Júlio de Mesquita Filho"—UNESP, Bauru, São Paulo, Brasil

Corresponding author
Aline Nonato de Sousa,
alinensousa1@ibb.unesp.br,
alinensousa1@gmail.com

## ABSTRACT

This is the first study to examine how different environmental factors may influence the distribution of swimming crab *Achelous spinimanus* across geographically distant and distinct habitats. We analyzed the influence of bottom water temperature and salinity, sediment texture and organic matter content on the spatiotemporal distribution of *A. spinimanus*. The crabs were collected from January 1998 until December 1999 by trawling with a shrimp fishing boat outfitted with double-rig nets. The sampling took place in Ubatumirim (UBM), Ubatuba (UBA) and Mar Virado (MV) bays, located in the northern coast of São Paulo State (Brazil). These three bays were chosen as they differed in many physiographic features. We captured 1,911 crabs (UBM = 351; UBA = 1,509; MV = 51), and there were significant differences in abundance between bays and between stations. The distribution of *A. spinimanus* was primarily associated with sediment features: abundance was higher in stations with sandy sediments classified as gravel, very coarse sand, and intermediate sand. Portunoidea usually burrow in the sediment for protection against predators and to facilitate the capture of fast prey. In addition, the station with the highest abundance of *A. spinimanus* was also naturally protected from fishing activities, and composed of heterogeneous sediment, in terms of grain size. Hence, the combination of a favorable sediment heterogeneity and protection from fishery activities seemed to be effective modulators of the abundance and distribution of *A. spinimanus* in these bays.

# INTRODUCTION

The abundance and distribution of organisms in the environment, as a rule, vary according to the variation of environmental resources (*Organista et al., 2005*). In this sense, *Organista*

*et al. (2005)* assumed that individuals may tolerate a wide variation of a given environmental factor (eurytopic) or not (stenotopic). Even slight variations in environmental conditions might lead to different behavioral, morphological, and physiological responses (*Thompson, 1991*). Moreover, predator–prey relationships and intra- or interspecific competition may also alter the seasonal distribution of different species (*Pinheiro, Fransozo & Negreiros-Fransozo, 1996*).

Studies on the distribution of benthic organisms have shown the importance of environmental factors such as temperature and salinity, sediment texture, and organic matter (*Abelló, Valladares & Castellón, 1988*; *Fariña, Freire & González-Gurriarán, 1997*; *Cartes et al., 2007*; *Bertini, Fransozo & Negreiros-Fransozo, 2010*; *Fransozo et al., 2016*; *Costa et al., 2016*).

According to *Mahiques (1995)*, the southeastern Brazilian coast, and especially within the northern part of São Paulo State, is characterized by many bays and an irregular landscape. In this region, the coastal line is very irregular, mainly due to the closeness to mountains of the Serra do Mar complex. These characteristics favor the establishment and development of marine organisms and, consequently, lead to a high biodiversity in such areas (*Negreiros-Fransozo et al., 1991*).

Due to its high productivity, the Ubatuba region is commonly exploited by shrimp fishing industry (*Mantelatto et al., 2016*) targeting mainly pink shrimp *Farfantepenaeus brasiliensis* (Latreille, 1817) and *F. paulensis* (Pérez-Farfante, 1967), and sea bob shrimp *Xiphopenaeus kroyeri* (Heller, 1862) (*D'incao, Valentini & Rodrigues, 2002*). Fishing by trawling is considered a destructive method for the benthic communities (*Branco & Fracasso, 2004*), and it often leads to a decrease in fishing stocks. When more profitable species become scarce, fishing fleets search for alternative resources. This is the case of *Achelous spinimanus* (Latreille, 1819) which has become a new target of the fishing fleets (*Santos, Negreiros-Fransozo & Padovani, 1995*; *Branco, Lunardon-Branco & Souto, 2002*; *Ripoli et al., 2007*), given its body size and particular taste, which are favorable to human consumption.

Besides its economic potential, *A. spinimanus*, as well as other swimming crabs, play a fundamental role in the trophic web of coastal ecosystems, being predators of various invertebrate groups (*Branco & Verani, 1997*). In the northern coast of São Paulo State, the representatives of Portunoidea are abundant and more diverse than other brachyurans (*Braga et al., 2005*; *Bertini, Fransozo & Negreiros-Fransozo, 2010*). In this region, several studies have investigated the relationship between these crustaceans and environmental variables. For instance, *Santos, Negreiros-Fransozo & Fransozo (1994)* and *Pinheiro, Fransozo & Negreiros-Fransozo (1996)* observed that the distributions of *A. spinimanus* and *Arenaeus cribarius* (Lamarck, 1818), respectively, were associated with sediment texture. *Pinheiro, Fransozo & Negreiros-Fransozo (1997)*; *Santos (2000)*, *Chacur & Negreiros-Fransozo (2001)*, *Andrade et al. (2013)*; *Andrade et al. (2014)*, *Lima et al. (2014)*, *Martins et al. (2014)*, and *Antunes et al. (2015)* related the distribution of Portunoidea with bottom temperature.

Considering that *A. spinimanus* is a new fishing resource, more studies on its biology are necessary to establish management strategies aiming at a more controlled fishing

and sustainable use. Although *Santos, Negreiros-Fransozo & Fransozo (1994)* and *Santos (2000)* determined the environmental factors influencing the distribution patterns of *A. spinimanus* in Fortaleza Bay (Ubatuba, SP), there have been no comparative studies addressing its distribution patterns in different areas, with different environmental features, at the same time. Aiming to fill this gap, we compared the distribution of *A. spinimanus* in three bays having distinct physiographical features. This comparison provided information on *A. spinimanus* distribution over a greater range of environmental conditions, expanding the knowledge from previous works conducted in only one bay. Furthermore, our goal was to indicate, by means of innovative and robust statistical analyses possible relationships between the spatiotemporal distribution of *A. spinimanus* and bottom water temperature and salinity, and sediment texture and organic matter content.

## MATERIAL & METHODS

### Study area

Ubatuba is located in the northern coast of São Paulo State, Brazil. This region has a unique geological conformation and is known for its very irregular coast (*Ab'Saber, 1955*). Ubatuba is influenced by three water masses: Coastal Water (CW: temperature $\geq$ 20 °C; salinity $\leq$ 36), Tropical Water (TW: temperature $\geq$ 20 °C; salinity $\geq$ 36), and South Atlantic Central Water (SACW: temperature $\leq$ 18 °C; salinity $\leq$ 36) (*Castro-Filho, Miranda & Myao, 1987*; *Odebrecht & Castello, 2001*; *De Léo & Pires-Vanin, 2006*). During late spring and early summer, the SACW penetrates into the coast's bottom layer and forms a thermocline over the inner shelf at depths of 10–15 m (*Castro-Filho, Miranda & Myao, 1987*; *Odebrecht & Castello, 2001*; *De Léo & Pires-Vanin, 2006*). During winter, the SACW retreats to the shelf break and is replaced by the CW, resulting in the absence of temperature stratification over the inner shelf during winter (*Pires, 1992*; *Pires-Vanin & Matsuura, 1993*).

For this study we chose three bays in Ubatuba that have distinct physiographical features such as shapes and outfall directions: Ubatumirim, Ubatuba, and Mar Virado (Fig. 1). Ubatumirim Bay (UBM) has an outfall heading southwest, and many islands and marine rock banks (Prumirim and Porcos Pequenos Islands facing its entrance, and Couves Island) (*Bertini, Fransozo & Negreiros-Fransozo, 2010*). Ubatuba Bay (UBA) has an east-facing outfall and a seaward constriction formed by rocky projections forming a more shallow inner area and a deeper outer area (>10 m deep) (*Mahiques, 1995*). Four rivers influence the sediment organic matter content in this bay (*Cetesb , 1996*), especially during rainy seasons, when larger amounts of sewage from the city of Ubatuba outflow into the area. Mar Virado Bay (MV) has a large outflow that faces southwest, with the Mar Virado Island at the eastern side of the bay entrance. The predominant substratum verified in this area comes from the sediment of two rivers, the Lagoinha and Maranduba rivers (*Mahiques, 1995*).

Since October 8, 2008, UBM, UBA and MV are parts of a Marine Protection Area (MPA) (APA Marinha do Litoral—Cunhambebe Sector) created by the Ministry of Environment (decree number 53.525). This MPA was established to ensure the conservation and sustainable use of marine resources. Fishing is only permitted for the subsistence of
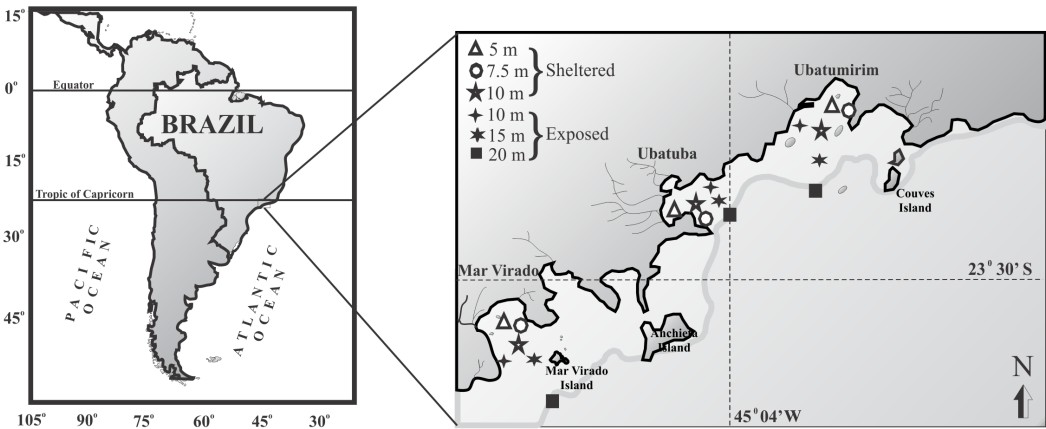

**Figure 1** Map of the Ubatuba region (northeastern coast of São Paulo State), Brazil, showing the three bays and their respective sampling stations (adapted from *Fransozo et al., 2013*).

traditional communities, by amateurs, and as a leisure activity, thus, commercial fishing is not allowed. These restrictions aim to protect the area and promote the rational use of its natural resources, ensuring the region's sustainable development.

## Sampling

We captured the swimming crabs monthly from January 1998 through December 1999. In each bay, six sampling stations were established: three stations were located in areas sheltered from the waves (5, 7.5 and 10 m deep), and three were located in exposed areas (10, 15 and 20 m deep) (Fig. 1). The stations (except the 7.5 and 10 m depths) were positioned along transects set parallel to the coastline. These stations were selected according to the following characteristics: their position relative to the bay's mouth, the presence of rocky shores or beaches along the bay's perimeter, freshwater inflow, proximity to offshore water, depth, and sediment texture.

Trawling was conducted on a commercial shrimp fishing boat outfitted with double-rig nets. Each area (18 sites) was trawled monthly (24 months) for 30 min and covered a total of 18,000 m$^2$ per trawl. Individuals were identified to species level (*Melo, 1996*) and sorted by sex, based on abdominal morphological features (male = triangular-shaped abdomen; female = round-shaped abdomen), and number of pleopods (males = two pairs; females = four pairs). The animals were collected within the guidelines of the *ICMBio - Instituto Chico Mendes de Conservação da Biodiversidade* - Ministério do Meio Ambiente (Permit no. 44329715).

At each station we took bottom and surface water samples using a Nansen bottle, and measured salinity (‰) and temperature (°C), using an optical refractometer and a mercury thermometer, respectively. Sediment samples were taken using a Van Veen grab, from which we obtained sediment texture and organic matter content. Depth was assessed with an echometer connected to a Global Positioning System (GPS). Immediately after collection, we put the sediment samples into labelled plastic bags and froze them to minimize the organic matter decomposition until further analyses.

Sediment analyses followed *Hakanson & Jansson (1983)* and *Tucker (1988)*. Two 50 g subsamples were taken, to which we added 250 ml of NaOH (0.2 N) to obtain the silt-clay fraction. We washed the subsamples using a sieve (0.063 mm mesh), washing away the silt-clay. The remaining sediment was dried and then differentially sieved, classifying the sediment grains according to the *Wentworth (1922)* scale.

Phi ($\varphi$) values were calculated based on the equation $phi = -\log_2 d$, where $d$ = grain diameter (mm), thus obtaining the following classes: -1|—0 (very coarse sand), 0|—1 (coarse sand), 1|—2 (intermediate sand), 2|—3 (fine sand), 3|—4 (very fine sand) e > 4 (silt-clay). Based on these values we calculated the central trend measurements, determining the most frequent granulometric fractions in the sediment. We calculated these values based on data graphically taken from cumulative sediment samples frequency distribution curves. We used values corresponding to the 16th, 50th and 84th percentages to determine the average diameter (AD), using the equation $AD = (\varphi_{16} + \varphi_{50} + \varphi_{84}/3)$ (*Suguio, 1973*).

To determine the sediment organic matter content we put 10 g subsamples in porcelain containers, previously labelled and weighed. They were oven-dried (500 °C for 3 h) and weighed. The difference between the initial and final weight indicated the organic matter content of each sampling station, what was later converted into percentages.

## Data analyses

Our data were not normally distributed (Shapiro–Wilk, $p > 0.05$) or homoscedastic (Levene's test, $p > 0.05$) (*Shapiro & Wilk, 1965*). Therefore, environmental factors (BT, bottom water temperature; ST, surface water temperature; BS, bottom water salinity; %OM, percentage organic matter; and Phi, sediment texture) were compared between years using a Mann–Whitney test (significance level = 5%). We compared BT, ST, BS, %OM and Phi values between bays, sampling stations, and seasons (Summer: January to March, and so on) using a Kruskal–Wallis test, followed by Dunn's post-hoc test (*Kruskal & Wallis, 1952*). Comparisons of the total abundance between bays and between stations were carried out using a Kruskal–Wallis test, followed by a post-hoc Dunn test (significance level = 5%).

We used a Redundancy Analysis (RDA) to detect possible relationships between the abundance of *A. spinimanus* and the environmental variables. This analysis requires the existence of, at least, two dependent variables. For that reason, we grouped the individuals into males (M) and females (F). The RDA produces final coordination scores that summarize the linear relationship between the explanatory and response variables. Only environmental variables with scores higher than 0.4 and lower than $-0.4$ were considered as biologically significant (*Rakocinski, Lyczkowski-Shultz & Richardson, 1996*). This analysis was performed using the Vegan package for R (*R Development Core Team, 2013*).

## RESULTS

The mean ST, BT and BS did not differ significantly between bays (Kruskal & Wallis, $p > 0.05$). However, significant differences in ST, BT and BS were observed between years and areas (Table 1).

**Table 1   Mean values (±standard deviation) for the environmental factors (ST, BT and BS) and results from the Kruskal–Wallis and Mann–Whitney test.**

|  |  | ST ± SD | BT ± SD | BS ± SD |
|---|---|---|---|---|
| Bays | Ubatumirim | $25.0 \pm 2.9$ | $22.9 \pm 2.8$ | $34.8 \pm 1.3$ |
|  | Ubatuba | $25.0 \pm 2.9$ | $23.2 \pm 2.8$ | $34.8 \pm 1.4$ |
|  | Mar Virado | $24.6 \pm 2.9$ | $22.8 \pm 2.8$ | $34.2 \pm 1.5$ |
|  | Test value /p | $H = 1.806/p = 0.405$ | $H = 1.529/p = 0.465$ | $H = 0.602/p = 0.740$ |
| Area | Sheltered | $25.2 \pm 2.9$ | $23.7 \pm 2.7$ | $34.4 \pm 1.5$ |
|  | Exposed | $24.6 \pm 2.9$ | $22.3 \pm 2.8$ | $34.8 \pm 1.3$ |
|  | Test value /p | $U = 20{,}095.5/p = 0.010$ | $U = 16{,}823.0/p < 0.001$ | $U = 18{,}339.0/p < 0.001$ |
| Year | 1998 | $25.2 \pm 2.6$ | $23.3 \pm 2.4$ | $34.7 \pm 1.4$ |
|  | 1999 | $24.6 \pm 3.2$ | $22.6 \pm 3.2$ | $34.5 \pm 1.5$ |
|  | Test value /p | $U = 19{,}915.5/p < 0.001$ | $U = 19{,}067.0/p < 0.001$ | $U = 20{,}550.0/p < 0.001$ |

**Notes.**

BT, Bottom water temperature; ST, Surface water temperature; BS, Bottom water salinity; $H$, Kruskal–Wallis test; $U$, Mann–Whitney test.

The highest variation in BT and ST, in all bays, was seen in summer and spring 1998/1999 (Fig. 2). In all bays, at the stations in exposed areas (10, 15 and 20 m deep) we observed clear differences between ST and BT (thermocline), especially in spring 1999 (Fig. 2). During autumn and winter, neither ST nor BT varied with depth (Fig. 2).

Temporally, the highest BS values were recorded in summer and autumn 1998, while in 1999, the highest BS was observed only in autumn (Fig. 3). The highest BS values were recorded in the 20 m deep station (Fig. 3). Significant variations in BS were recorded only in 1999.

The presence of three water masses typically observed in the studied region is noticeable in the BS and BT diagrams. From these diagrams it can be said that CW was prevalent in both years and in all bays, while the influence of SACW and TW was noticed only in 1999 (Figs. 4 and 5).

Grain size and organic matter content differed between bays and between sites (Fig. 6). There was a gradual increase in phi values from north to south, with mean phi values of 3.8, 4.4 and 5.5 in UBM, UBA and MV, respectively. The highest %OM was recorded in UBA (5.9%), followed by MV (4.5%) and UBM (3.6%). The 20 m depth had the lowest %OM average value (3.3%) ($p < 0.05$) (phi = 3.0). With respect to the 10 m deep stations, we observed the highest %OM (6.2%) and mean phi (5.3), i.e., %OM increased as the sediment grain size decreased.

We captured 1,911 *A. spinimanus* individuals: 1,255 in 1998, and 656 in 1999. The highest abundance was recorded in UBA (1,509), followed by UBM (351) and MV (51). The abundance varied throughout the seasons, being the highest in fall and spring 1998 in UBA (Table 2). When comparing the abundance between bays, we observed that MV had significantly lower abundance ($p < 0.01$). In UBM, the highest abundance recorded in 1998 was in the 20 m deep station, while in 1999, it was in the 15 m station. In UBA, the highest abundances in 1998 and 1999 were recorded in the 7.5 m deep station. In MV, even though we did not observe significant differences in the abundance of swimming

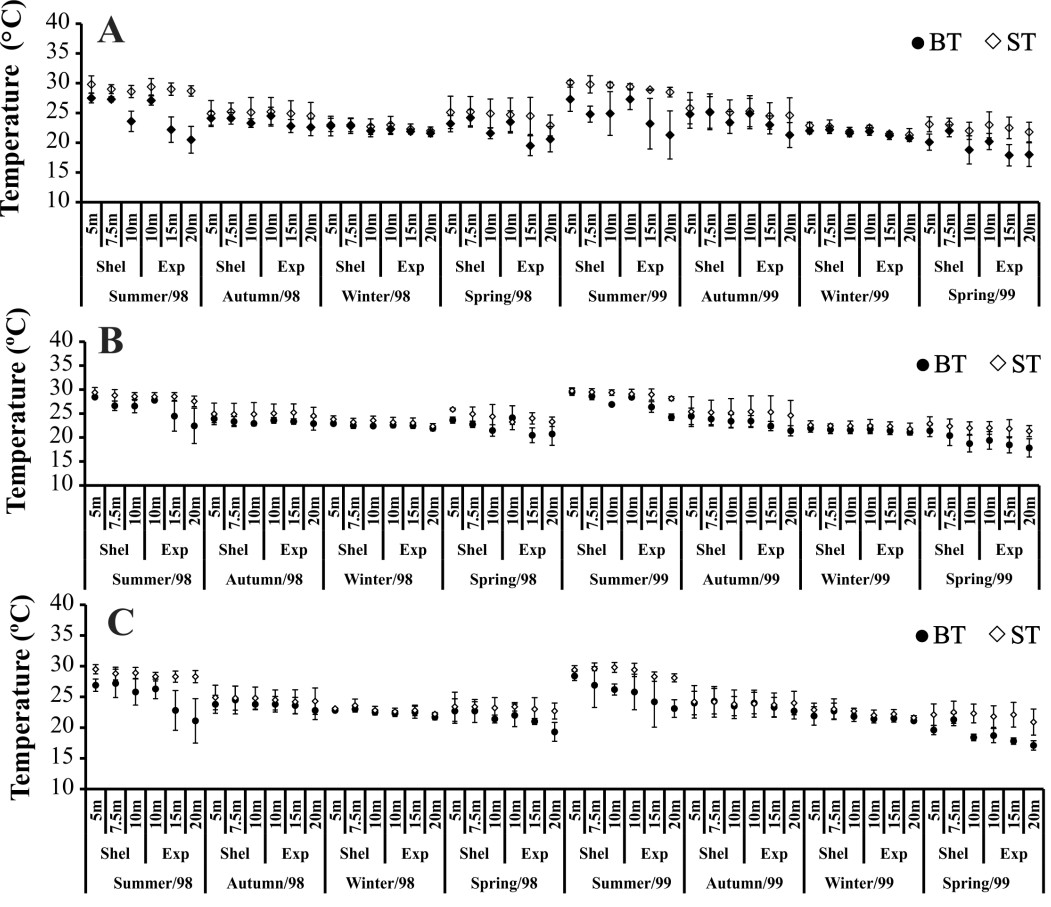

**Figure 2** Bottom and surface temperature variation per seasons, areas and stations in 1998 and 1999. (A) Ubatumirim Bay. (B) Ubatuba Bay. (C) Mar Virado Bay. (BT) Bottom temperature. (ST) Surface temperature

crabs between stations ($p > 0.05$), individuals were found in only two stations (7.5 and 20 m deep) (Fig. 7).

The results of the axis 1 in the RDA (which explained 92.3% of the variation in our data) revealed that the sediment features (%OM and Phi) were factors that mostly affected the individuals' distribution in all studied bays, for both years (Table 3). Based on this analysis the abundance of *A. spinimanus* is inversely proportional to phi value. The highest swimming crab abundances were seen in stations with low and intermediate phi values, i.e., with heterogeneous sediment, mainly composed of gravel, very coarse sand, coarse sand and medium sand.

## DISCUSSION

Based on the water temperature and salinity recorded in this study, we can infer that the Coastal Water current prevailed in the three bays. This water mass is characterized by salinity under 36 and temperature higher than 20 °C (*Castro-Filho, Miranda & Myao,*

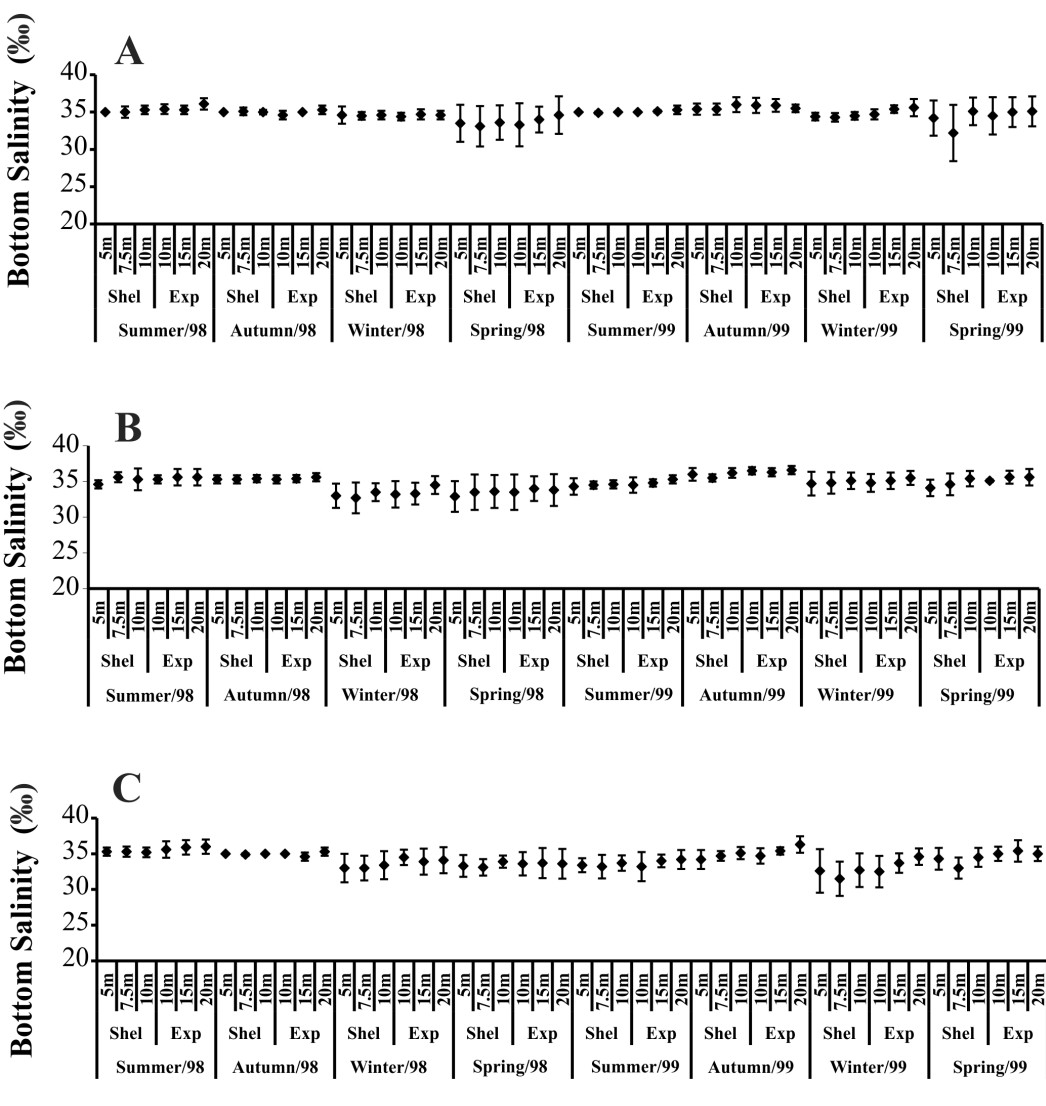

**Figure 3** **Bottom salinity variation per seasons, areas and stations in 1998 and 1999.** (A) Ubatumirim Bay. (B) Ubatuba Bay. (C) Mar Virado Bay.

*1987*). The effects of the South Atlantic Central Waters and Tropical Water masses could only be noticed in the second year, 1999. The SACW is a cold water mass, with temperatures under 18 °C and salinity lower than 36, which reaches the deepest layers of the coastal water column and generates a thermocline (*Pires, 1992*). In this study, this thermocline was more evident in exposed areas, especially during the spring 1999.

Several studies have reported the influence of SACW and its physicochemical characteristics over the temporal abundance of decapod crustaceans along the southeastern Brazilian coast (*Furlan et al., 2013*; *Bochini et al., 2014*; *Andrade et al., 2014*; *Castilho et al., 2015*). In this study, we speculate that this water mass has a negative influence on the abundance of *A. spinimanus*, as its decreased in summer and spring 1999 in UBA. UBA Bay seems to be more vulnerable than UBM and MV bays with respect to the effects of oceanic

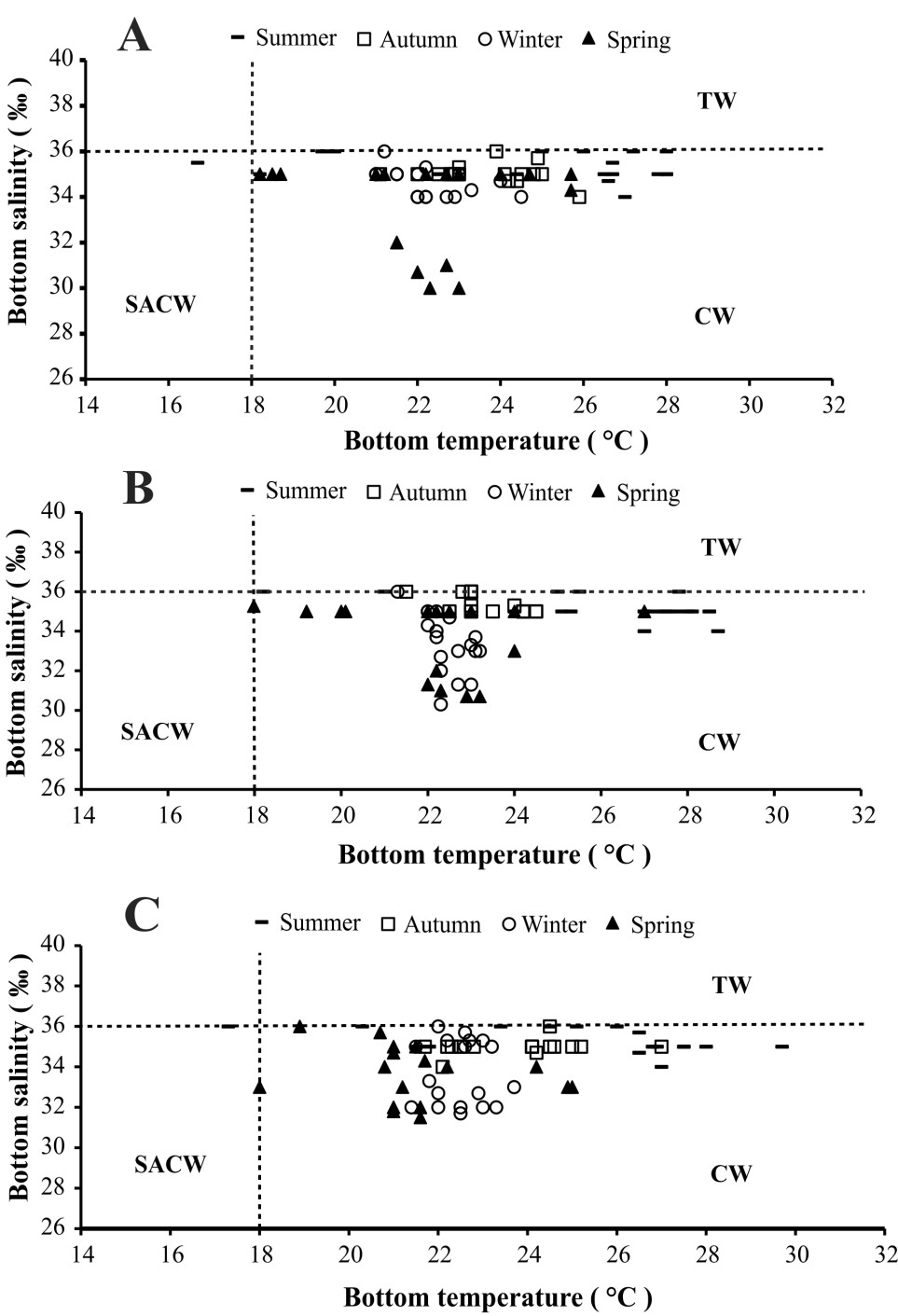

**Figure 4** Diagram showing the seasonal variation of water temperature and salinity from January 1998 to December 1998, at Ubatumirim, Ubatuba and Mar Virado, São Paulo State littoral, southeastern coast of Brazil. (A) Ubatumirim Bay. (B) Ubatuba Bay. (C) Mar Virado Bay. (CW) Coastal Water. (TW) Tropical Water. (SACW) South Atlantic Central Water.

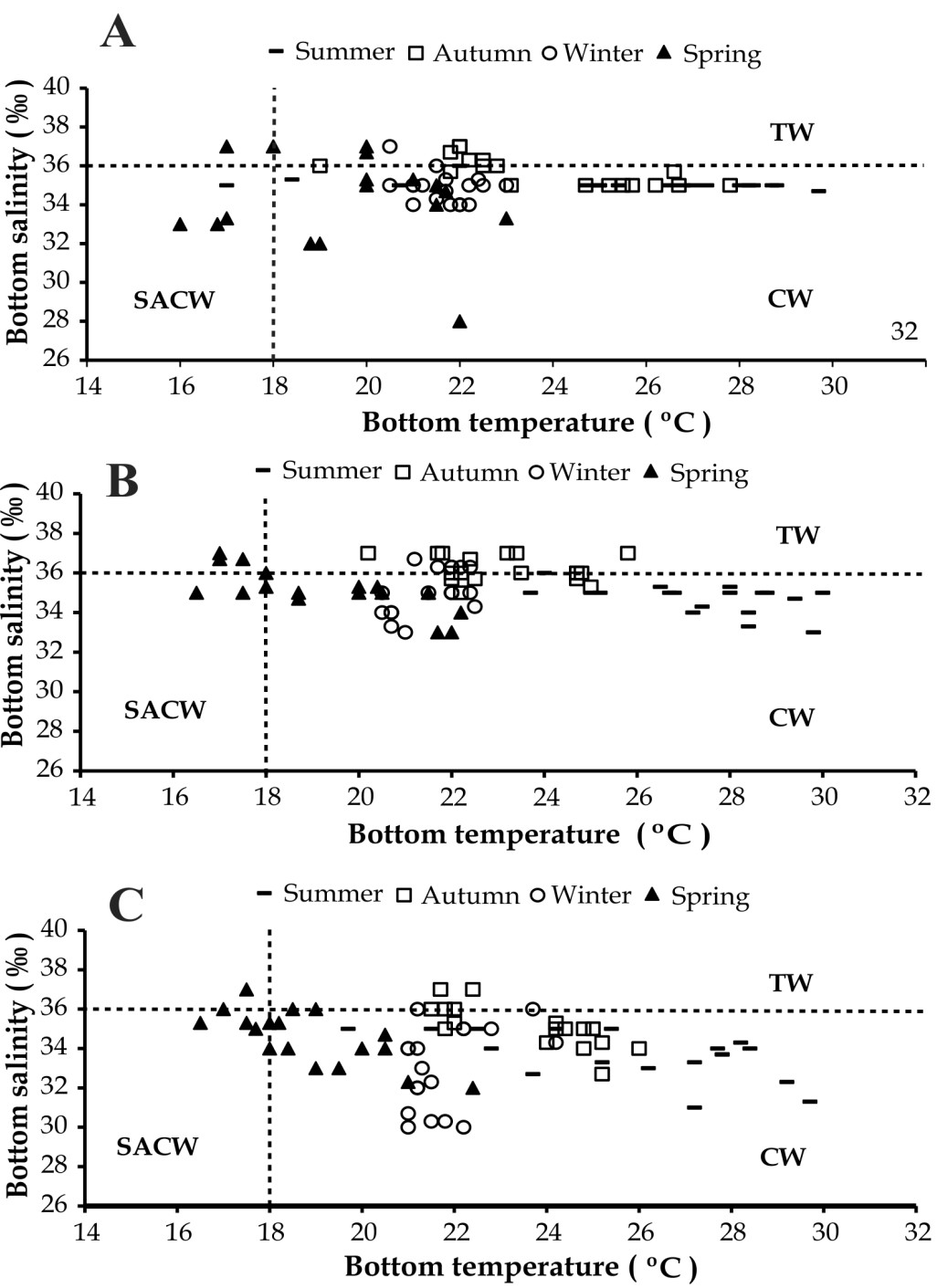

**Figure 5  Diagram showing the seasonal variation of water temperature and salinity from January 1999 to December 1999, at Ubatumirim, Ubatuba and Mar Virado, São Paulo State littoral, southeastern coast of Brazil.** (A) Ubatumirim Bay. (B) Ubatuba Bay. (C) Mar Virado Bay. (CW) Coastal Water. (TW) Tropical Water. (SACW) South Atlantic Central Water.

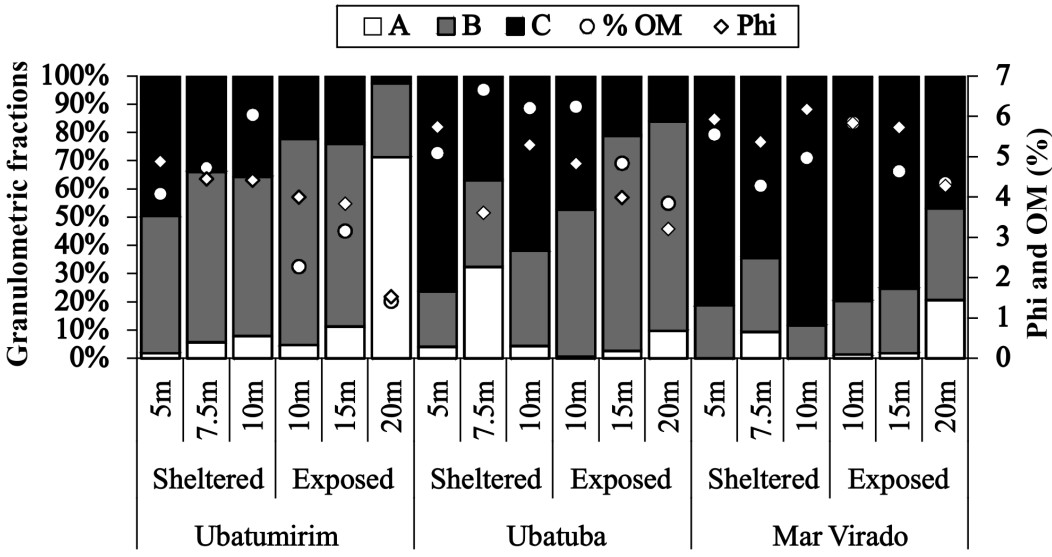

**Figure 6 Proportions of grain-size classes, central tendency of bottom sediments (phi) and mean values of the organic matter content of the sediment (% OM) for each depth in Ubatumirim, Ubatuba and Mar Virado, São Paulo State littoral, southeastern Brazil.** (A) Class A (gravel, very coarse sand, coarse sand, and intermediate sand). (B) Class B (fine and very fine sand). (C) Class C (silt-clay).

currents as it has no physical protection, such as islands nearby. The coastline geography and the presence of islands seen along the northern coast of São Paulo can attenuate the incidence of cold currents on the coast (*Mahiques, 1995*). Thus, as a consequence, the effects of SACW in UBA Bay are stronger, and may explain why in 1999 in UBA *A. spinimanus* migrated towards sheltered areas. *Mantelatto & Fransozo (2000)*, studying the same bay, showed that from September 1995 through August 1996, individuals of *A. spinimanus* were grouped in a more sheltered site located in the inner portion of Ubatuba Bay (which was not included in this study).

Temperature is widely accepted as a limiting factor in the distribution of marine organisms (*Lewis & Roer, 1988*), since many metabolic and physiological processes in crustaceans (such as molting, growth, and oocyte maturation) depend on this variable (*Sastry, 1983*). Previous studies (*Santos, Negreiros-Fransozo & Fransozo, 1994*; *Santos, 2000*; *Bertini & Fransozo, 2004*; *Lima et al., 2014*), carried out in the same area as the present one, have also seen evidenced of the bottom water temperature's influence on the biology of *A. spinimanus*.

In UBM Bay, the stations at 15−20 m deep were the only ones with sediment composed of higher granulometric fractions. Thus, even though these sediments were exposed to the effects of water masses, they were favorable to the establishment of *A. spinimanus*. As well, in the same bay, at the same time as when this study was performed, it was possible to see a higher abundance of *Callinectes danae* Smith, 1869 in the shallower sampling sites (*Antunes et al., 2015*). Since *Shinozaki-Mendes, Manghi & Lessa (2012)* state that *C. danae* displays agonistic and territoriality behaviors, we speculate that this behavior may have hampered the establishment of *A. spinimanus* in the same area. The abundance of one species in a

Table 2 *Achelous spinimanus* (Latreille, 1819). Number of individuals in each month and station sampled, indicating the results from the Dunn test (Columns with one same capital letter in common did not differ statistically, $p > 0.05$).

| Season | Month | Sheltered area | | | Exposed area | | | Total | | Sheltered area | | | Exposed area | | | Total | |
|---|---|---|---|---|---|---|---|---|---|---|---|---|---|---|---|---|---|
| | | 5 m | 7.5 m | 10 m | 10 m | 15 m | 20 m | Month | Season | 5 m | 7.5 m | 10 m | 10 m | 15 m | 20 m | Month | Season |
| | | | | | | | | **Ubatumirim** | | | | | | | | | |
| | | | | | | **1998** | | | | | | | | **1999** | | | |
| | Jan | 0 | 6 | 3 | 0 | 8 | 3 | 20 | | 1 | 1 | 0 | 1 | 6 | 2 | 11 | |
| Summer | Feb | 0 | 10 | 2 | 1 | 9 | 2 | 24 | 56 | 0 | 0 | 3 | 0 | 30 | 7 | 40 | 61 |
| | Mar | 0 | 1 | 1 | 1 | 0 | 9 | 12 | | 0 | 0 | 6 | 0 | 1 | 3 | 10 | |
| | Apr | 0 | 0 | 1 | 0 | 1 | 8 | 10 | | 0 | 0 | 1 | 3 | 29 | 1 | 34 | |
| Autumn | May | 0 | 2 | 0 | 0 | 7 | 1 | 10 | 37 | 0 | 0 | 0 | 0 | 7 | 4 | 11 | 54 |
| | Jun | 0 | 10 | 0 | 0 | 2 | 5 | 17 | | 0 | 1 | 0 | 0 | 3 | 5 | 9 | |
| | Jul | 0 | 5 | 0 | 1 | 0 | 1 | 7 | | 0 | 0 | 0 | 1 | 3 | 3 | 7 | |
| Winter | Aug | 0 | 0 | 0 | 0 | 5 | 8 | 13 | 28 | 0 | 0 | 0 | 0 | 6 | 7 | 13 | 26 |
| | Sep | 0 | 2 | 0 | 0 | 2 | 4 | 8 | | 0 | 0 | 0 | 0 | 4 | 2 | 6 | |
| | Oct | 0 | 0 | 0 | 0 | 3 | 23 | 26 | | 0 | 0 | 0 | 0 | 6 | 20 | 26 | |
| Spring | Nov | 0 | 2 | 0 | 2 | 4 | 2 | 10 | 39 | 0 | 1 | 1 | 4 | 3 | 3 | 12 | 50 |
| | Dec | 0 | 1 | 0 | 0 | 2 | 0 | 3 | | 0 | 1 | 0 | 0 | 6 | 5 | 12 | |
| **Total** | | 0 | 39 | 7 | 5 | 43 | 66 | 160 | 160 | 1 | 4 | 11 | 9 | 104 | 62 | 191 | 191 |
| **Duun Test** | | A | BC | AB | AB | BC | C | | | A | A | A | A | B | B | | |
| | | | | | | | | **Ubatuba** | | | | | | | | | |
| | Jan | 1 | 7 | 0 | 1 | 1 | 0 | 10 | | 5 | 9 | 0 | 1 | 0 | 10 | 25 | |
| Summer | Feb | 1 | 30 | 0 | 0 | 1 | 31 | 63 | 138 | 0 | 0 | 1 | 2 | 3 | 12 | 18 | 69 |
| | Mar | 0 | 56 | 1 | 0 | 0 | 8 | 65 | | 0 | 11 | 0 | 0 | 0 | 15 | 26 | |
| | Apr | 0 | 129 | 0 | 0 | 1 | 8 | 138 | | 1 | 52 | 0 | 0 | 0 | 5 | 58 | |
| Autumn | May | 0 | 127 | 0 | 0 | 0 | 7 | 134 | 496 | 0 | 35 | 1 | 0 | 0 | 1 | 37 | 112 |
| | Jun | 1 | 206 | 0 | 0 | 1 | 16 | 224 | | 0 | 8 | 0 | 0 | 0 | 9 | 17 | |
| | Jul | 0 | 42 | 0 | 0 | 0 | 4 | 46 | | 2 | 28 | 0 | 0 | 1 | 2 | 33 | |
| Winter | Aug | 0 | 24 | 0 | 0 | 0 | 13 | 37 | 100 | 0 | 44 | 0 | 0 | 0 | 6 | 50 | 125 |
| | Sep | 0 | 14 | 0 | 0 | 0 | 3 | 17 | | 0 | 39 | 0 | 0 | 0 | 3 | 42 | |
| | Oct | 0 | 31 | 0 | 0 | 1 | 12 | 44 | | 2 | 11 | 0 | 0 | 2 | 8 | 23 | |
| Spring | Nov | 0 | 26 | 0 | 0 | 2 | 8 | 36 | 324 | 0 | 25 | 0 | 0 | 1 | 11 | 37 | 145 |
| | Dec | 2 | 234 | 0 | 0 | 1 | 7 | 244 | | 0 | 74 | 0 | 0 | 0 | 11 | 85 | |
| **Total** | | 5 | 926 | 1 | 1 | 8 | 117 | 1,058 | 1,058 | 10 | 336 | 2 | 3 | 7 | 93 | 451 | 451 |
| **Duun Test** | | A | B | A | A | AC | BC | | | A | B | A | A | A | B | | |
| | | | | | | | | **Mar Virado** | | | | | | | | | |
| | Jan | 0 | 6 | 0 | 0 | 0 | 16 | 22 | | 0 | 1 | 0 | 0 | 1 | 1 | 3 | |
| Summer | Feb | 0 | 3 | 0 | 0 | 0 | 7 | 10 | 32 | 0 | 0 | 0 | 0 | 0 | 1 | 1 | 9 |
| | Mar | 0 | 0 | 0 | 0 | 0 | 0 | 0 | | 0 | 0 | 0 | 0 | 0 | 5 | 5 | |
| | Apr | 0 | 0 | 0 | 0 | 0 | 0 | 0 | | 0 | 0 | 0 | 0 | 0 | 2 | 2 | |
| Autumn | May | 0 | 0 | 0 | 0 | 0 | 0 | 0 | 1 | 0 | 0 | 0 | 0 | 0 | 3 | 3 | 5 |
| | Jun | 0 | 1 | 0 | 0 | 0 | 0 | 1 | | 0 | 0 | 0 | 0 | 0 | 0 | 0 | |

**Table 2** (*continued*)

| Season | Month | Sheltered area | | | Exposed area | | | Total | | Sheltered area | | | Exposed area | | | Total | |
|---|---|---|---|---|---|---|---|---|---|---|---|---|---|---|---|---|---|
| | | 5 m | 7.5 m | 10 m | 10 m | 15 m | 20 m | Month | Season | 5 m | 7.5 m | 10 m | 10 m | 15 m | 20 m | Month | Season |
| | Jul | 0 | 0 | 0 | 0 | 0 | 1 | 1 | | 0 | 0 | 0 | 0 | 0 | 0 | 0 | |
| Winter | Aug | 0 | 0 | 0 | 0 | 0 | 0 | 0 | 2 | 0 | 0 | 0 | 0 | 0 | 0 | 0 | 0 |
| | Sep | 0 | 1 | 0 | 0 | 0 | 0 | 1 | | 0 | 0 | 0 | 0 | 0 | 0 | 0 | |
| | Oct | 0 | 1 | 0 | 0 | 0 | 0 | 1 | | 0 | 0 | 0 | 0 | 0 | 0 | 0 | |
| Spring | Nov | 0 | 0 | 0 | 0 | 0 | 0 | 0 | 2 | 0 | 0 | 0 | 0 | 0 | 0 | 0 | 0 |
| | Dec | 0 | 1 | 0 | 0 | 0 | 0 | 1 | | 0 | 0 | 0 | 0 | 0 | 0 | 0 | |
| **Total** | | 0 | 13 | 0 | 0 | 0 | 24 | 37 | 37 | 0 | 1 | 0 | 0 | 1 | 12 | 14 | 14 |
| **Duun Test** | | Non-significant *p*-value for the Kruskal–Wallis test | | | | | | | | | | | | | | | |

**Table 3** *Achelous spinimanus* (**Latreille, 1819**). Results from the redundancy analysis (RDA): ordination of the first two canonical axes, with environmental variable data and demographic categories' abundance from Ubatumirim, Ubatuba and Mar Virado. Coefficients greater than or equal to +0.4 or lower than or equal to 0.4 were considered ecologically relevant (see *Rakocinski, Lyczkowski-Shultz & Richardson, 1996*) and are shown in bold.

| | Axis 1 | Axis 2 |
|---|---|---|
| Eigenvalue | 0.020 | 0.001 |
| % of Variance | 0.923 | 0.076 |
| **Demographic categories** | | |
| Males | −0.329 | −0.159 |
| Females | −0.554 | 0.094 |
| **Environmental variables** | | |
| Bottom temperature | 0.040 | −0.958 |
| Bottom salinity | 0.080 | 0.139 |
| Organic matter | **−0.418** | −0.450 |
| Phi | **0.685** | −0.523 |

certain place may be considered an ecological response through its adaptations to both the environmental factors and intra- and interspecific interactive processes (*Shinozaki-Mendes, Manghi & Lessa, 2012*).

This study indicates the texture of the sediment as the main factor modulating the distribution of *A. spinimanus*. In all sampled bays, *A. spinimanus* abundance was higher in the stations composed mainly of heterogeneous sediment. This may be explained by the foraging and refuge options created by the heterogeneous sediments: as the more heterogeneous the sediments are, the more microhabitats there are (*Bertini, Fransozo & Melo, 2004*). Previous studies (*Santos, Negreiros-Fransozo & Fransozo, 1994*; *Bertini & Fransozo, 2004*; *Furlan et al., 2013*, for instance) have also described a higher abundance of *A. spinimanus* in areas where the sediment texture was more heterogeneous.

The highest abundance of *A. spinimanus* was recorded in UBA (78.9% of the total number of individuals) in the 7.5 m deep station, which may be explained by the heterogeneous texture of the sediment and a high percentage of organic matter of this station. According to *Moore (1958)*, areas with finer sediment grains may have a higher percentage of organic matter when compared to the ones with coarser grains. However, we observed an association

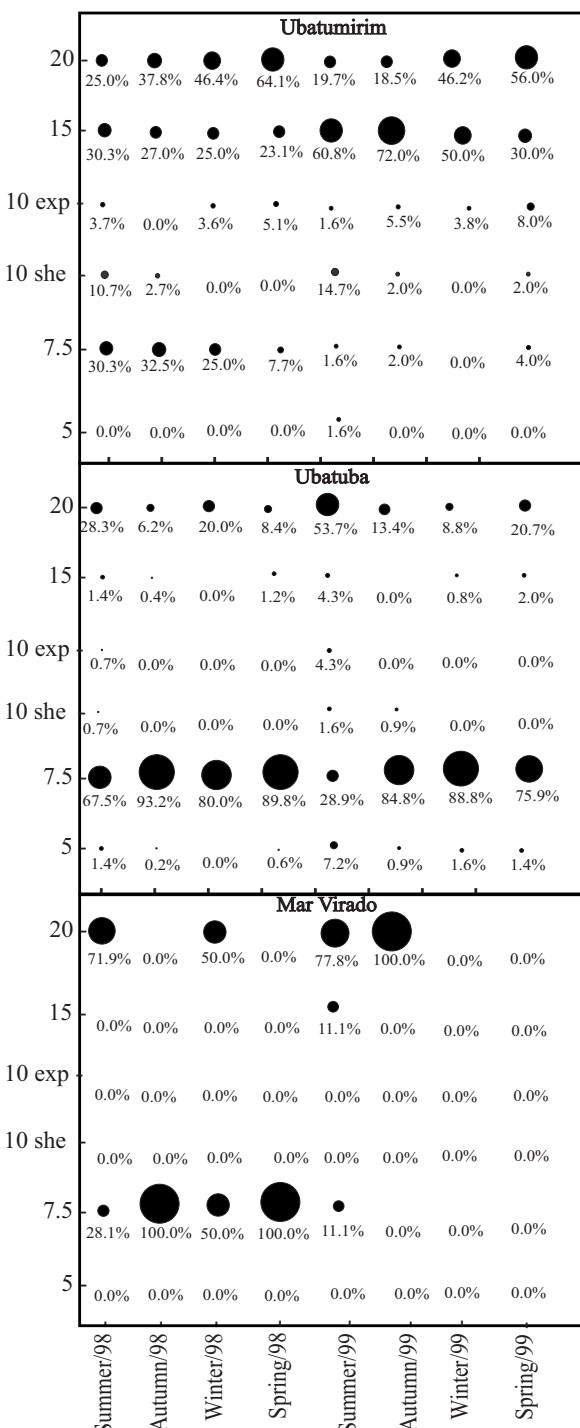

**Figure 7** *Achelous spinimanus* (**Latreille, 1819**). Percentage of individuals in each season in the different bays and in each depth. Circle sizes are proportional to the percentages of individuals. (Exp) exposed. (She) sheltered.

between the higher granulometric fractions and the organic matter content levels, even though organic content was also positively associated with the silt-clay fraction. The higher organic matter content observed in the 7.5 m deep station is related to gravel-composed sediments, which are of biogenic nature, comprised of the remains of mollusk shells, crustacean carapaces and echinoderms.

*Lima et al. (2014)* studying *A. spinimanus* in UBA in 2000 found fewer crabs (402 individuals) than in any year examined in the present study, although they used the same sample effort as the present study. This is probably due to those authors did not sample in the 7.5-m deep sampling station. One may thus assume that sheltered areas with heterogeneous sediments may provide a favorable habitat to the establishment of *A. spinimanus*. Moreover, this station is naturally protected from fisheries as it has many natural obstacles such as rocks and coral fragments, which damage fishing gears. Thus, the lower fishing pressure could have contributed to the higher abundance of individuals there. Lower fishing pressure can promote habitat complexity, favoring the establishment of individuals (*Kaiser et al., 2002*; *Fransozo et al., 2016*).

*Mantelatto et al. (2016)* indicated that UBA stations with high diversity indexes are naturally protected against fishing due to the difficulty of carrying out trawls in the area, which consequently provides less exposure of the local benthic fauna to the actions caused by trawls. According to *Kaiser et al. (2002)*, the impacts of trawling on ecosystems include the reduction of habitat complexity, and changes in species abundance and distribution patterns and overall benthic community structure. Furthermore, *Fransozo et al. (2016)* describes trawling as destructive and destabilizing to benthic communities, since it is not selective (i.e., does not capture only the fishery's target) and jumbles the seafloor, displacing or removing many other organisms from their natural environments.

The low abundance observed in MV can be related to the sediment features observed in that bay. This bay's sediment is composed mainly of silt-clay brought from the continent, as well as a consequence of the physical barriers formed mainly by the São Sebastião Channel, together with Anchieta and Vitória islands. Accordingly, *Santos, Negreiros-Fransozo & Fransozo (1994)* sampled 126 individuals of *A. spinimanus* in Fortaleza Bay (November 1988 through October 1989), whereas *Hiroki (2012)*, 20 years later (November 2008 through October 2009), and using the same sampling procedure, collected only 5 individuals. It is noteworthy that besides observing a lower abundance, *Hiroki (2012)* also observed a decrease of the higher granulometric fractions (gravel, very coarse sand, coarse sand and intermediate sand).

Based on our investigation, we highlight the role that environmental factors such as the sediment texture play in the establishment and development of *A. spinimanus* populations. Portunoidea usually burrow in the sediment for protection against predators or to facilitate the capture of fast prey (*Schöne, 1961*) such as fishes. Muddy sediments, however, make burrowing and the intake of water for gas exchange more difficult.

According to *McNaughton & Wolf (1970)*, the dominance of certain species in a given habitat may be explained mainly by two opposite hypotheses: (1) the dominant species are generalists and adapted to a wide variation in environmental conditions and therefore, are not limited by them; or (2) the dominant species are specialists and are well adapted

to one or some aspects of their habitat. In the study carried out by *Bertini, Fransozo & Negreiros-Fransozo (2010)*, some species seemed to be generalists and not restricted to a certain type of substrate (e.g., *Callinectes ornatus* Ordway, 1968 and *Hepatus pudibundus* (Herbst, 1785)), while others were frequently associated with specific sediment types (e.g., *Libinia ferreirae* Brito Capello, 1871 and *A. spinimanus*). Despite its higher abundance in coarser sediments, *A. spinimanus* cannot be characterized as a stenotopic species until it is shown that the this crab is limited by a number of characteristics/ parameters.

## CONCLUSION

Overall, this study broadens the knowledge on the sediment features most favorable to the establishment and development of *A. spinimanus* populations. Moreover, it provides a basis for comparison with current data, and it also attests the efficiency of the implemented strategies in 2008–2009 for the species conservation.

## ACKNOWLEDGEMENTS

We thank NEBECC co-workers for their help during the fieldwork and Dr. Maria Lucia Negreiros-Fransozo for her constructive comments and valuable grammar review of this manuscript.

### Funding
The work was supported by the São Paulo Research Foundation (FAPESP) (grant numbers 94/4878–8, 97/12108–6, 97/12106–3, 97/ 12107–0 and 98/3134–6). The funders had no role in study design, data collection and analysis, decision to publish, or preparation of the manuscript.

### Grant Disclosures
The following grant information was disclosed by the authors:
São Paulo Research Foundation (FAPESP): 94/4878–8, 97/12108–6, 97/12106–3, 97/ 12107–0, 98/3134–6.

### Competing Interests
The authors declare there are no competing interests.

### Author Contributions
- Aline Nonato de Sousa and Thiago Elias da Silva analyzed the data, contributed reagents/materials/analysis tools, prepared figures and/or tables, authored or reviewed drafts of the paper, approved the final draft.
- Giovana Bertini, Rogério Caetano da Costa and Adilson Fransozo conceived and designed the experiments, performed the experiments, contributed reagents/materials/analysis tools, prepared figures and/or tables, authored or reviewed drafts of the paper, approved the final draft.

- Fabiano Gazzi Taddei contributed reagents/materials/analysis tools, prepared figures and/or tables, authored or reviewed drafts of the paper, approved the final draft.

## Field Study Permissions

The following information was supplied relating to field study approvals (i.e., approving body and any reference numbers):

Field experiments were approved by the Instituto Chico Mendes de Conservação da Biodiversidade - ICMBio of the Ministério do Meio Ambiente - MMA.

## Data Availability

The raw data are provided in a Supplemental File and Table 2.

## Supplemental Information

Supplemental information for this article can be found online at http://dx.doi.org/10.7717/peerj.5720#supplemental-information.

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
