# Peer review of "Modulating factors of the abundance and distribution of Achelous spinimanus (Latreille, 1819) (Decapoda, Portunoidea), a fishery resource, in Southeastern Brazil"

_PeerJ, doi:10.7717/peerj.5720_

## Round 0.1 · original submission · Major Revisions

Please, note that you have to consider all suggestions made by the reviewers and not in a cosmetic way. Your manuscript needs serious work before resubmission. One of the reviewers explicitly asked for a deep revision of the language. I am not also a native English speaker and I know how hard is to write in another language, but this is the rule. Therefore, try hard to improve this in the next version.

Please, also note, that reviewer #3 is specifically asking for the raw R codes. He/she feels that this is needed in order to him/her continue the review process.

You have a good chance to have your manuscript approved IF you really consider suggestions and comments made by the reviewers.

Reviewer 1 ·

Basic reporting

The manuscript titled "Modulating factors on the abundance and distribution of Achelous spinimanus (Latreille, 1819): a fishery resource in the Brazilian Littoral Southeastern" analyzes the influence of abiotic variables on the distribution of the species in three bays of southeastern Brazil.
The work presents technical-scientific merit and is well elaborated, however, it presents some items that need adjustments, which are identified throughout the PDF and below.

Experimental design

The content discussed has already been analyzed and published by other authors, with notoriously original items, only a combination of some elements (as highlighted by the authors on lines 79-84).
There are two items that need to be clarified and / or better analyzed:
1. The authors use the identification of three different water bodies (Coastal Water, Tropical Water and South Atlantic Central Water) based on a 1987 publication. With so many changes in global temperature, has there been no values ​​of these currents? What about the seasonal variation? Still, there is a temperature gap (from 18o to 20o) and salinity (36) that does not belong to any of these currents. The study of physical oceanography is more complex than a combination of temperature and salinity.
2. The influence of abiotic variables occurs together. It is known that it is impossible to quantify all the variables that influence animal life, but the variables that have been measured must be analyzed together and separate factors must be separated. Better explaining: if there are 18 distinct sites and 2 years, authors can not group all sites and evaluate by year, as each site is different and it is not allowed to group separate elements to be analyzed together. In the same way that it is not possible to affirm that the granulometry, by itself, determines the distribution of the animals, since there is also the influence of the water masses, the different bays, the depth, the organic matter, etc. That is, we suggest that analyzes be performed in which all variables are considered, and not isolate them.

Validity of the findings

The authors provided only the spreadsheet with abiotic data, and this made it impossible for us to evaluate some biological aspects. In this context, the main inconsistency of the work is not to insert data essential for the understanding of the distribution, which are the sizes, sexes and maturity of the copies. It is well known that for most Portunidae there is age and sexual segregation and different uses of Habitat. It is possible that the granulometry is directly associated with the juvenile or adult phase of the specimens; or not. The reader finishes the manuscript without having the knowledge of this information that is fundamental, since the subject approaches the factors that modulate the abundance and distribution.

Additional comments

In general, the main failure of the work is to analyze the variables independently, when there is notoriously a set of influences. For example, in Table 1, only the years were compared. But is there a difference between the sheltered areas of the waves and exposed areas? And between the bays? And between the months?
In the discussion, the authors make the same mistake, of analyzing each variable separately. How can it be said, for example, that temperature influences distribution if there are also other variables? So it is not the temperature, in fact, it is a set of factors. In this case, it is up to the authors to quantify the influence and importance of each variable. For this, we suggest a multiple analysis, a generalized linear model or a multivariate one so that the variables are analyzed in a weighted way.
Another item that was deficient in the work is that the authors affirm that the individuals were sexed, which is a fundamental aspect in the study of distribution. However, this information was not explored by location and time of year. We believe that this information is indispensable, as well as the classification of young people and adults.

Annotated reviews are not available for download in order to protect the identity of reviewers who chose to remain anonymous.

·

Basic reporting

The authors have sought assistance to improve language and grammar. However, there remains room for improvement, particularly with tense and word choice. The need for improvement to a level of professional English makes reading difficult in many places in the manuscript. For example, see lines 79-82. Also, what is a “naturally excluded” fishing area? What is meant by predatory fishing gear? Are not all fishing gear operated by predators (i.e., humans) using active (trawls) and passive (drift net, long lines) fishing gear?

The Introduction and background describe the context of this study. The references are relevant, but there is room for improvement as to how they are cited and referenced in the text. This issue relates more to proper English usage than a problem in not referencing in the text. For example, line 37 refers to “those authors” which refers to a previously cited paper. I suggest instead that the reference be repeated.

The overall structure conforms to Peer J standards. However, all of the cited references are in italics which is not correct. They should be Times New Roman, no italics.

The tables and figures are relevant, although more will be said later on what is presented. All are of good quality except Figure 5. The resolution of this figure needs improvement.

Raw data is provided. However, if the authors wish to reference it, then the data needs to be cited in the text.

Experimental design

Original research is presented that is within the scope of Peer J.

The research question is well defined, but the knowledge gap that it attempts to fill is not. No justification is presented for repeating a study of environmental parameters affecting the spatiotemporal distribution of A. spinimanus. Previous studies with overlapping research goals and study areas are referenced, so the authors are aware of them ( for examples, Lima et al 2014, Andrade et al. 2014). Within the lines of 85 – 90, a clear justification for repeating research and how this new effort attempts to fill knowledge gaps left by previous studies needs to be presented.

The description and justification of the sampling design needs improvement, particularly how sample locations were selected within bays. If Fig. 1 is an accurate representation of the three bays, then how the 20 m depth sample locations are justified as being located within their respective bays needs to be explained. They do not appear to be within any of the bays. The same can be said for the 15 m locations at MV and UBM.

The methods for species identification, temperature and salinity recording, and sediment analysis are described in sufficient detail.

Validity of the findings

The relationship between sediment composition and abundance of A. spinimanus confirms how sediment type influences the distribution of crabs previously shown by Lima et al. 2014. In addition, crabs are most abundant in “naturally excluded” area of UBA (this location needs to be named in the abstract). What is not addressed is how the RDS analysis is overwhelmingly influenced by the high abundance of crabs at UBA. Is not the sediment type relationship with crab abundance highly influenced by most crabs being caught in a naturally excluded fishing area? If there had been a bay that was not “naturally excluded” where similar numbers of crabs were found, the findings would be more convincing. In this respect, the discussion of stenotopic versus eutytopic is not convincing, especially since more than one environmental parameter is needed to define a species as such. More needs to be said to explain why a swimming crab is so influenced by sediment type.

The temperature data is pooled to make comparisons between years. Instead, since a rationale of the experimental design is to compare bays, the temperatures within each bay for each year should be shown and statistically compared. When reported in the results, the analysis should be in one paragraph instead of the serval now written between lines 166 – 183. The tables need to be better referenced, such as between lines 171 and 175 where not table is cited. Somewhere in the text is should be stated that the data passed or di not pass the Shapiro-Wilk and Levene’s Tests.

The potential interaction between Callinectes danae and A. spinimaus is interesting, but pure speculation and needs to be stated as such until someone studies the interactions between these two species.

The discussion of the influence of SASW is very interesting. What it needs is more information about the local oceanography of the coast and bays before any relationship between spatiotemporal distribution and season can be made. Otherwise, what is said between lines 227 and 232 is speculation and needs to be stated as such if included in a revised manuscript. The aspect of the bays (a measure currently named outfall) most certainly will influence the affect of SACW, although coastal currents will modulate that effect.

In conclusion, a reanalysis of temperature data and local oceanography will help support the explanations for the spatiotemporal distribution of Achelous spinimanus. Differences and similarities between this study and previous ones showing sediment-animal relationship for A. spinimanus need to be clearly reported and discussed. Priority needs to be given to the finding that crabs were most abundant where they were protected from harvesting and how that has a lesser or greater effect on distribution than sediment type. Until it is shown that the requirements of this crab species are narrow for a number of resources/parameters, any discussion about steno/eurytopic characterization should be very limited.

Additional comments

This is an interesting paper with good potential for publication. Personally, I think more might be said about how areas naturally protected from commercial harvesting are important for maintaining populations of this species. I also hope more can be added about local oceanography and SACW and how they might influence the distribution of Acheles spinimanus.

·

Basic reporting

This paper is interesting and relevant to the field. The authors could make some minor improvements in the English, but overall it is very well written. I enjoyed reading the literature review.

However, I note that I am unable to comment on the impact of the findings because the authors did not include their raw code. I would be happy to continue the review once I see the R code used to compute the results outlined in this paper.

Experimental design

I am unable to comment on the impact of the findings because the authors did not include their raw R code.

Validity of the findings

I am unable to comment on the impact of the findings because the authors did not include their raw R code.

Additional comments

I would like to continue reviewing this paper, but I need to see the R code used to compute the results outlined here. If the authors could please include their .R file, I will continue my review.

---

## Round 0.2 · Minor Revisions

The authors have accepted and incorporated major suggestions made by the reviewers. However, there are still some room for improvements. Please see the remaining suggestions made by the reviewers. We are close to acceptance now! Keep going...

·

Basic reporting

This revision shows a wonderful improvement in professional English. I commend the authors on their diligence and work. Now more clearly shown, there is sufficient background to set up the authors' hypotheses. The literature is amply supports the background, methods, and ideas discussed. Figures and tables all show professional structure and raw data are shared (at with reviewers -- not clear if they will be supplied as a supplement since not referenced as so anywhere in the text). I suggest that the authors cite their raw data as supplementary materials where that is appropriate in the text.

There are still a few places needing clarity and I have uploaded an annotated copy of the manuscript for the authors.

Experimental design

This revision meets all four criteria.

Validity of the findings

This revision meets all four criteria. The authors have made good progress revising their ideas and taking the reviewer suggestions. As a result, the ideas are more clearly understood.

Additional comments

This revision is greatly improved and addresses my concerns with the first draft. I have uploaded an annotated copy of this manuscript in the hope that my few suggestions will be seen as helpful and instructive.

·

Basic reporting

I thank the authors for their timely resubmission of their manuscript materials. I have several comments in descending order of importance:

1. I strongly urge the authors to employ the assistance of a native English speaker to help modify the grammar and verbiage of the paper. The paper cannot be published until the English grammar is corrected.

2. I appreciate the excellent references and structure of the introduction to the paper.

3. The figures and tables are sufficient and pertinent to the main points of the paper. I thank the authors for sharing the raw R code used for their analyses, and I request that for the next submission, the R code and associated data tables are submitted as .R and .txt (or .csv or .xlsx) files, respectively.

Experimental design

The authors make clear the novelty of their research. I would like the authors to comment on the use of 20 year old data to build these models. I expect that there may have been significant ecosystem shifts in this period of time, and wonder how this could affect the findings.

Validity of the findings

The conclusions are outlined thoroughly in the discussion point, but again, I would like commentary about how these results may change if updated survey data were incorporated into the models. I would also like more commentary on the applicability of these findings to resource managers.

Additional comments

Thank you for making some revisions and including your code. Please see my comments above and within the document.

---

## Round 0.3 · Minor Revisions

I have gone over your manuscript, and while the science is fine, I agree with previous reviewers that the English still needs editing. I have edited the paper in detail myself (see attached PDF document and the journal office will send you the original Word doc), but there are a couple areas where you need to think about what to write, and also ensure the English is perfect.

I look forward to seeing a revised version of your work.

---

## Round 0.4 · accepted · Accept

The new version has been revised well, and I am happy to move this into production.

#